# Comorbidities, Complications and Non-Pharmacologic Treatment in Idiopathic Pulmonary Fibrosis

**DOI:** 10.3390/medsci6030059

**Published:** 2018-07-24

**Authors:** Paloma Millan-Billi, Candela Serra, Ana Alonso Leon, Diego Castillo

**Affiliations:** Department of Respiratory Medicine, Hospital de la Santa Creu I Sant Pau, C/Sant Antoni M. Claret, 167, 08025 Barcelona, Spain; PMillan@santpau.cat (P.M.-B.); CSerraX@santpau.cat (C.S.); AAlonso@santpau.cat (A.A.L.)

**Keywords:** idiopathic pulmonary fibrosis, palliative care, comorbidities, pulmonary rehabilitation, ambulatory oxygen therapy

## Abstract

Idiopathic pulmonary fibrosis (IPF) is a chronic, progressive and fatal disease. The treatment is challenging and nowadays a comprehensive approach based not only in pharmacological strategies is necessary. Identification and control of comorbidities, non-pharmacological treatment, prevention and management of exacerbations as well as other areas of care (social, psychological) are fundamental for a holistic management of IPF. Gastroesophageal reflux, pulmonary hypertension, obstructive sleep apnea, combined with emphysema, lung cancer and cardiovascular involvement are the main comorbidities associated with IPF. Non-pharmacological treatment includes the use of oxygen in patients with rest or nocturnal hypoxemia and other support therapies such as non-invasive ventilation or even a high-flow nasal cannula to improve dyspnea. In some patients, lung transplant should be considered as this enhances survival. Pulmonary rehabilitation can add benefits in outcomes such control of dyspnea, exercise capacity distance and, overall, improve the quality of life; therefore it should be considered in patients with IPF. Also, multidisciplinary palliative care programs could help with symptom control and psychological support, with the aim of maintaining quality of life during the whole process of the disease. This review intends to provide clear information to help those involved in IPF follow up to improve patients’ daily care.

## 1. Introduction

Idiopathic pulmonary fibrosis (IPF) is a progressive and unpredictable disease with a poor 5-year survival outcome and great impact on patient’s quality of life. Therefore, improving survival has been the main objective of research studies and the goal of IPF’s treatment. In the last decade, scientific evidence changed the perspective of the disease by demonstrating the efficacy of two drugs in slowing down the loss of lung capacity in patients with IPF.

Despite of these promising new treatments, IPF remains an incurable disease with a complex comprehensive approach. In this respect, optimal management may include comorbidity identification and treatment, as well as non-pharmacological interventions such as ambulatory oxygen, pulmonary rehabilitation and lung transplant.

Besides, during the past century life expectancy has improved significantly, especially in developed countries. In these societies, quality of life (QOL) has become an important issue, particularly in older ages with significant comorbidities. This population is well represented by IPF patients. So, QOL is also a goal of the management plan. Nonetheless, palliative care plays a very important role as one of the main keys in preserving QOL and alleviating symptoms.

In the current article, we will review the most recent medical literature about IPF comorbidities, complications and non-pharmacological treatment in order to provide clear information to help those involved in IPF follow up to improve patients’ daily care.

## 2. Comorbidities

Management of comorbidities in IPF is very important due to their significant impact in life expectancy and quality of life. However, due to the few well-designed clinical trials in this field quality data is scarce, and most recommendations are based in expert opinions or single centers experiences. Figure 1 summarizes the main evidence discuss below.

### 2.1. Pulmonary Hypertension

Pulmonary hypertension (PH) is defined as a mean pulmonary artery pressure ≥25 mmHg evidenced on right heart catheterization—the gold standard procedure for diagnosis [1]—although this is not regularly performed in patients with idiopathic pulmonary fibrosis (IPF). The prevalence of PH at the initial phase of IPF ranges from 8–17%, while in patients on a lung-transplant registry it ranges from 30–50%. PH may complicate the clinical course of IPF and has a prognostic impact [2].

PH should be suspected in patients with evidence of right heart failure on physical exam, dyspnea or oxygen desaturation out of proportion for their pulmonary function, disproportionately low diffusion capacity, or evidence of pulmonary artery enlargement or right ventricular hypertrophy on imaging studies. Transthoracic echocardiogram is the study used to screen for PH and right ventricular systolic pressure >35 mmHg has a sensitivity of >85% for detecting PH in patients with IPF [3].

Besides, PH could be aggravated by other comorbidities, such as obstructive sleep apnea (OSA), pulmonary thromboembolic disease, emphysema or cardiological abnormalities (ventricular diastolic dysfunction). Therefore, it is important to screen for these conditions and treat them if necessary.

No effective therapy has been established for PH associated with IPF. In ATS/ERS/JRS/ALAT 2015 guidelines [4] authors made a conditional recommendation against the use of Sildenafilo, a phosphodiesterase-5 inhibitor, based on a phase III randomized study (Sildenafil Trial of Exercise Performance in Idiopathic Pulmonary Fibrosis [STEP-IPF] [5] which showed no significant benefit on mortality or acute exacerbations in a pooled analysis. In the same way, the 2015 guidelines mentioned previously made a conditional recommendation against the use of Bosentan or Macitentan, both dual endothelin receptor antagonists, showed no significant benefit in mortality although the data suggested an improvement in the composite outcome of mortality and disease progression.

Despite this results, nitric oxide pathway treatment showed a possible benefit as some secondary outcomes were positive [6]. Trials with ambrisentan and riociguat were stopped after showing no benefit or even an increased risk of death and other serious adverse events in the intervention groups. Based on these data, current guidelines do not recommend use of selective pulmonary vasodilators for the treatment of PH in IPF [7,8].

### 2.2. Emphysema

Combined pulmonary fibrosis and emphysema (CPFE) prevalence is around 30% in IPF patients. There is still controversy about the amount of emphysema observed in high-resolution computed tomography to define CPFE, and it is proposed ≥10% emphysematous involvement throughout the lungs. Because of the distinct clinical, functional, radiological, and pathological characteristics of CPFE, it has been proposed as a phenotype of IPF or as a separate clinical entity [3].

Individuals with CPFE are usually male with an extensive smoking history and increased oxygen requirement. The pulmonary function test often shows relatively preserved total lung capacity and forced vital capacity (FVC), with a disproportionate reduction in the diffusion capacity of the lung for carbon monoxide (DLCO). The incidence of pulmonary hypertension (50%) and lung cancer (47%) is higher in this group, with an increased risk of death in some cohorts [9].

General treatment recommendations in these patients includes smoking cessation, inhaled bronchodilators according to chronic obstructive pulmonary disease guidelines, oxygen therapy (if there is an indication), rehabilitation, vaccines and lung-transplant evaluation. Both Pirfenidone (ASCEND) and Nintedanib (INPULSIS 1–2) trials included patients with emphysema. Despite the inclusion of emphysema patients, all the variability of patients with CPFE was not represented in those trials because patients with airway obstruction and lower DLCO were excluded. With the aforementioned trials, the efficiency of these antifibrotic drugs was maintained in the subgroup of patients with emphysema [10].

### 2.3. Obstructive Sleep Apnea 

The prevalence of OSA in IPF patients has been reported from 10–88% and is often undiagnosed. In the Lancaster et al. [11] study, most of the patients had moderate to severe OSA (apnea-hypopnea index (AHI) >15 events per hour), without strong correlation between AHI and body mass index.

The presence of OSA was associated with a more rapid clinical deterioration independent of the severity of IPF. OSA with prolonged sleep-related desaturation was associated with worse prognosis, both in terms of mortality and clinical deterioration [12].

There are several theories to explain the relation between OSA and IPF. It has been suggested than the main contributor is the impairment of the upper airway stability due to decreased lung volumes, especially during the REM (rapid eye movement) phase. This can facilitate the upper airway collapse. On the other hand, intermittent hypoxia, produced in OSA, is a potent stimulus to oxidative stress causing systemic inflammation and tissue damage that could carry out progression of pulmonary fibrosis [13].

The initiation of continuous positive airway pressure (CPAP) has shown to improve daily activities and quality of life measures [1]. Also, it could reduce the incidence of ischemic heart disease and nocturnal hypoxemia, which is strongly correlated with PH [9]. There are currently prospective randomized clinical trials evaluating the treatment of OSA in IPF that hopefully will give us more data about the impact of this treatment in the prognosis of the disease.

### 2.4. Gastroesophageal Reflux

Gastroesophageal reflux (GER) is highly prevalent in patients with IPF (87–94%), with almost half of the patients being asymptomatic. GER and microaspiration have been proposed as risk factors for the development and progression of IPF [1]. Both acid and non-acid reflux have been suggested as potential factors for worsening IPF progression. In a retrospective analysis [14] of three randomized controlled trials (Evaluating the Effectiveness of Prednisone, Azathioprine, and N-acetylcysteine in Patients With Idiopathic Pulmonary Fibrosis [PANTHER] [15], Anticoagulant Effectiveness in Idiopathic Pulmonary Fibrosis [ACE] [16], STEP [5]) anti-acid therapy was associated with lower decline in FVC and fewer acute exacerbations of IPF (AE-IPF), while in other retrospective analyses, ASCEND (Assessment of Pirfenidone to Confirm Efficacy and Safety in IPF) [17], CAPACITY 004 and 006 (Clinical Studies Assessing Pirfenidone in Idiopathic Pulmonary Fibrosis: Research of Efficacy and Safety Outcomes) [18] it was associated with an increased risk of infection [19]. Current IPF guidelines have proposed a conditional recommendation for anti-acid treatment based in previous findings, although scientific evidence is scarce in this topic [20]. For that reason, randomized prospective clinical trials are now warranted to further study this therapy in this population.

Recently, alkaline reflux has also been linked to alveolar injury in epithelial cells, playing a part in pathogenesis of IPF and also increasing the risk of exacerbations. Hypothesizing a potential benefit in controlling acid and non-acid GER, Raghu et al. [21] evaluated the impact of laparoscopic anti-reflux surgery (LARS) to control its effects in pulmonary lung function. This retrospective study, performed with few patients and no control group, showed a non-significant reduction in FVC deterioration, with no effect on mortality nor in surgical complications. This is the first surgical work attempting to lower GER in order to control IPF. A randomized multicenter controlled trial is now ongoing (WRAP-IPF: Weighing Risks and Benefits of Laparoscopic Anti-Reflux Surgery in Patients with Idiopathic Pulmonary Fibrosis) that could provide additional insights. A recent meta-analysis [22] compiled 13 studies in this field, confirming the importance of addressing GER medically and surgically in IPF management.

Finally, microaspiration has been proposed to have a role in the development of AE-IPF. The association between microaspiration and AE-IPF was strengthened in a recent study by Molyneaux et al. [23]. In this study, an increase in the copies of bacterial DNA in the bronchoalveolar lavages of patients experiencing AE-IPF was observed in contrast to stable IPF. Also, the study observed a change in the microbiota during exacerbations, with a higher relative abundance of two potentially pathogenic agents; *Stenotrophomonas* spp. and *Campylobacter* spp. The presence of *Campylobacter* spp., a gastrointestinal pathogen, in the respiratory microbiota is supposed to have arisen from silent microaspiration of gastric contents.

### 2.5. Cardiovascular Comorbidities

The importance of this comorbidity was highlighted in the study of Nathan et al. [24], as cardiovascular diseases in IPF patients were associated with a shorter survival compared to a similarly matched group with chronic obstructive pulmonary disease [1]. Additionally, in a recent study in Finland, ischemic heart disease is reported to be the second highest cause of death in IPF patients [25].

Coronary artery disease has an estimated prevalence of 3–68% in IPF patients, sharing several risk factors as male gender, increasing age and smoking history [2]. Other cardiovascular comorbidities that are described in IPF patient are arrhythmias, with a prevalence from 5 to 20%, and heart failure with prevalence from 4 to 26% [2]. The most commonly reported arrhythmias in IPF patients are atrial fibrillation and atrial flutter. The presence of hypoxia, pulmonary hypertension, chronic inflammation, coronary artery disease or heart failure association [26] could explain the increasing incidence of the arrhythmias.

Coronary disease could be increased by the promotion of atherosclerotic effects hypothesized of pulmonary fibrosis, mediated by increasing serum levels of interleukins, cytokines and circulating immune complexes. Interleukin-4 (IL-4), Interleukin-8 (IL-8) and tumor necrosis factor-α (TNF-α) are involved in angiogenesis and are also present in fibro-proliferative processes beyond the lungs. All these disorders might explain the possible mechanisms of increased atherogenesis [26].

Dyspnea not fully explained by the severity of IPF should be evaluated with an electrocardiogram, stress radionuclide myocardial perfusion imaging, or stress echocardiography should be considered [3].

Treatment of cardiovascular comorbidities in this group of patients do not differ from the general population.

### 2.6. Lung Cancer

The incidence of lung cancer in IPF patients ranges from 1–48%. IPF is an independent risk factor for lung cancer in addition to smoking [1]. Squamous cell carcinoma slightly predominates over adenocarcinoma in IPF patients. This last histologic type may arise from abnormally proliferating bronchioles in areas of honeycomb cyst, as suggested by Calio et al. [27].

Therapeutic modalities need to be determined in this group, because interventions could provoke exacerbations of IPF. A study in patients with non-small cell lung cancer (NSCLC) who underwent surgery showed that the survival to five years was 40%, referring to reduced early dead with wedge resection (caused by respiratory failure) and long-term better prognosis with lobectomy [28].

Acute exacerbation increased also with chemotherapy (especially docetaxel) and with radiotherapy [29]. Interstitial lung disease (ILD) was described as a risk factor for drug-induced lung injury, without having an optimal chemotherapy treatment to reduce this effect [1].

Pirfenidone was proposed to reduce acute exacerbations after surgery in the PEOPLE Study (“Efficacy and safety of perioperative pirfenidone for prevention of acute exacerbation of idiopathic pulmonary fibrosis in lung cancer patients undergoing pulmonary resection”). [30]. Furthermore, the anti-proliferative effects of pirfenidone and nintedanib may synergize with chemotherapeutic treatments, but additional data is needed [3].

### 2.7. Venous Thrombosis and Pulmonary Embolism

Reduced mobility with venous stasis and a probable involvement and activation of the coagulation cascade is involved in the pathogenesis of IPF, evidenced by elevated levels of tissue factor and fibrin; this increases the risk of venous thrombosis and, consequently, pulmonary embolism (PE) [31].

Epidemiological analysis reported a relationship between thromboembolic phenomenon and pulmonary fibrosis [32,33]. In fact, a review of death certificates in United States showed that venous thromboembolism (VTE) contributed to 1.74% of all IPF deaths, and the risk of VTE in IPF at the time of death was 34% higher than the risk of VTE in the background decedent population [34]. Also, in an American retrospective study, the incidence of PE reported in IPF patients was almost three times higher than the population without IPF [35]. Therefore, in patients with progression of symptoms and stable pulmonary function parameters PE should be excluded.

Considering this, Leuschner et al. [36] conducted a study to investigate the concordance of ventilation/perfusion (V/Q)-SPECT (Single Photon Emission Computed Tomography) and CT-angiography in patients with pulmonary fibrosis, clinical deterioration and suspicion of pulmonary embolism (PE). Twenty-two patients underwent both procedures. Of those 22 patients, PE was detected in two patients by both methods and in seven patients only by SPECT, concluding that more thromboembolic events were detected in patients with fibrosis by V/Q-SPECT. Despite these findings, it is important to remember that CT allows, in addition to the vascular study, to identify alterations in pulmonary parenchyma and fibrosis progression.

Regarding thromboembolic phenomenon management, prolonged treatment with anticoagulants is necessary. Both American and European guidelines [37,38] recommend at least three months of anticoagulant therapy, after which time the risk–benefit of continued treatment is necessary. This last point is important as the ACE-IPF study [16] showed that warfarin increased mortality in progressive IPF with no benefits of therapy in those patients who had no other indication for anticoagulation therapy.

### 2.8. Diabetes Mellitus

The 2011 guidelines for diagnosis and management of IPF [39] made a small mention of diabetes mellitus (DM) as a risk factor of IPF. Previously, Enomoto et al. [40] reported that DM could increase the risk for IPF, although no references regarding impact on survival were made.

In order to investigate the relationship between IPF and DM, a Chinese group performed a systematic review on this topic [41]. After reviewing 17 studies, authors found a wide range of prevalence of DM (9.67% to 56%) in IPF patients. An association between IPF and DM was considered consistent although the pathogenic mechanisms implicated remain unknown. It is clear that rapid detection and treatment of diabetes mellitus is important, as well as prospective studies to determine its influence on the mortality of these patients and establish a possible pathogenic relationship.

### 2.9. Hypothyroidism

The prevalence of hypothyroidism is 1–2% for men and 5–9% for women being the autoimmune thyroiditis the most common etiology [42]. In a case-control study conducted by Oldham et al. [43] a higher prevalence of hypothyroidism in patients with IPF was reported, with 13% of men and 28% of women affected. A link between hypothyroidism and IPF is not well established yet, but it is hypothesized that both may share a baseline autoimmune disorder.

### 2.10. Anxiety and Depression

Anxiety and depression are comorbidities commonly find in patients with interstitial lung diseases including IPF. The prevalence of anxiety ranges from 30–50% while the prevalence of depression from 20–30% [44,45]. Anxiety and depression presented in those patients is not directly related with physiological parameters although is well known that dyspnea and progression of the disease are aggravating factors. Also, it has a great impact on the quality of life of those patients.

Recently, Glaspole et al. [46] performed a study to evaluate the frequency of prolonged anxiety and depression among IPF patients. The results confirmed that dyspnea is a major contributor for both anxiety and depression. Also, on the multivariate analysis, use of oxygen was associated with anxiety. Another interesting finding was that only 25% of patients with anxiety or depression received treatment, suggesting a significant number of undiagnosed and/or untreated patients.

In the same line, a recent Cochrane review [47] highlighted the need for additional study of pharmacologic treatment of anxiety in chronic obstructive pulmonary disease (COPD), which could also be applied to the population with IPF.

In conclusion, there is still some doubt about the directionality of this relationship between IPF and symptoms of anxiety and depression. However, what is clear is that the need for pharmacological or behavioral therapy must be detected early and assessed individually in order to improve global management of IPF and quality of life.

## 3. Complications

### Acute Exacerbation

The main complication of IPF is the acute exacerbation, a condition that remains unpredictable and it is associated with a high risk of death. The continuous search for risk factors and clinical predictors for exacerbations is needed in order to improve the management of those patients.

In 2016 the international working group for AE-IPF reviewed the definition of acute exacerbation. The new definition describes any acute (less than a month), clinically significant respiratory deterioration (both idiopathic and triggered events) characterized by the presence of new bilateral ground-glass opacity and/or consolidation not fully explained by cardiac failure or fluid overload [48].

The most important risk factors for AE-IPF are low forced vital capacity, rapid decline in FVC (less than six months), high alveolar-to-arterial oxygen pressure difference, pulmonary hypertension at the time of evaluation for lung transplantation, gastroesophageal reflux and air pollution [1].

There is no proven effective treatment for AE-IPF. The 2011 IPF guidelines recommended supportive care and suggested that most patients could be treated with corticosteroids, without clear evidence to support this approach [39]. While receiving high doses of glucocorticoids, it seems rational to use empiric antibiotic therapy. A prospective randomized trial proposed procalcitonin-guided antibiotic treatment showing shorter duration of therapy. The results showed a similar mortality and days of ventilation compared to the standard clinician-determined antibiotic treatment. Another prospective study suggests usefulness of azithromycin therapy for AE-IPF [49]. On the other hand, some authors sustain that the use of high-dose steroids affects adversely the outcome of AE-IPF and propose a steroid avoidance strategy based on benefits obtained in an uncontrolled study [50].

Lung transplantation is proposed for appropriate patients with IPF, according to IPF guidelines. Transplant eligible patients should be referred to a transplantation center for early evaluation in the course of their disease, preferably before an episode of AE-IPF. Mechanical ventilation and extracorporeal membrane oxygenation are proposed as bridge for transplantation candidates. A more strict selection criteria for patients candidates to mechanical ventilation and the use of protective ventilation strategy could improve AE-IPF prognosis [49].

Moreover, treatment with nintedanib is proposed to reduce the risk of developing an acute exacerbation although no statistically significant difference in the incidence rates of investigator reported acute exacerbations in the nintedanib and placebo groups were seen [51]. The combined evidence from the TOMORROW (Efficacy of a Tyrosine Kinase Inhibitor in Idiopathic Pulmonary Fibrosis) and INPULSIS (Efficacy and safety of nintedanib in idiopathic pulmonary fibrosis) trials had showed a significant benefit in favor of nintedanib on time to first investigator-reported acute exacerbation, with a hazard ratio of 0.53 (95% confidence interval (CI):0.34–0.83; *p* = 0.0047) [52].

Pirfenidone could reduce the incidence of postoperative AE-IPF in patients with lung cancer and it´s safe to be use in these circumstances [30]. Besides, it is proposed to be used as an add-on therapy in AE-IPF-patients receiving corticosteroids to reduce inflammation, but more evidence is needed to confirm these findings [53].

Other proposed treatment is the recombinant human soluble thrombomodulin, which has an anti-inflammatory, anticoagulant and antifibrinolytic effect, that had favorable mortality rate in some uncontrolled studies and is now been test in a phase three clinical study [49].

Other promising treatments that are in study are hemoperfusion with polymyxin B-immobilized fiber column and therapies against autoantibodies (plasma exchange, rituximab and intravenous immunoglobulin).

Statins and anti-acid treatment were proposed to reduce incidence of AE-IPF, with prospective studies needed to confirm this hypothesis [14,54].

## 4. Non-Pharmacological Treatment

Current IPF treatment guidelines highlight the importance of non-pharmacological treatment in these patients. In the Figure 2 there is a summary of the main domains that the management plan should cover.

### 4.1. General Measures 

IPF is associated with increased dyspnea and therefore, with a hypermetabolic state and high resting oxygen consumption. It is not unusual to observe a reduction in body weight and loss of muscle mass that contributes to reduction in functional capacity and low physical activity. Therefore, nutritional counseling is very important in the global management of IPF patients. This is even more important in patients referred to for lung transplant.

In smokers it is very important to provide information about smoking cessation assistance. As other chronic lung diseases, smoking increase the risk of presenting with other severe comorbidities as lung cancer and co-existence of emphysema. So, quitting smoking is a cornerstone of management.

Regarding vaccination, no specific studies have been conducted in IPF patients. Although, as with other patients with chronic respiratory diseases, they have a higher risk of suffering from respiratory infections. Hence, the annual influenza and anti-pneumococcal vaccination is recommended in all patients.

### 4.2. Oxygen

Compared to other chronic respiratory diseases, exertional hypoxemia is a common finding in IPF. However, hypoxemia at rest usually occurs in the final stages of the disease, which means the subsequent indication for ambulatory oxygen. The prescription of oxygen may reduce breathlessness and increase physical capacity by improving the gas exchange. But also, ambulatory oxygen is helpful in the treatment of frequent comorbidities in IPF associated with poor prognosis such as pulmonary hypertension and right heart failure.

There is scarce evidence with no clinical randomized trials to evaluate the benefit of oxygen in interstitial lung disease, including IPF. Results of the low-quality evidence provided by retrospective studies are contradictory. For example, Douglas et al. [55] analyzed 487 patients with IPF, of which 133 were receiving oxygen therapy. The study did not show an improvement of survival in the treatment group. A recent systematic review showed no consisted effects of short term oxygen therapy on dyspnea during exertion in ILD although exercise capacity was increased [56]. In the discussion, the authors acknowledge that the results of all the studies analyzed are confusing as the patients with more severe and progressive disease were more likely to receive oxygen therapy. On the other hand, Egan et al. in a previous review of non-pharmacological treatment in IPF, consider oxygen therapy as a critical component of the management of IPF [57] despite the lack of evidence. Currently, there is an ongoing study to evaluate the effects of ambulatory oxygen during daily life on health status and breathlessness in fibrotic lung disease [58] that hopefully will provide more evidence for the use of oxygen in ILD patients.

Regarding exertional dyspnea, it is well known that exercise limitation and increase of dyspnea have an important impact on quality of life. In fact, hypoxia during the 6 min walking test (6-MWT) is an independent prognostic factor of mortality and disease progression [59,60,61]. Lettieri et al. [61] analyzed retrospectively 81 patients with IPF, establishing that those with greater desaturation and reduced meters walked presented a higher mortality. In this context, Hook et al. [62] analyzed 255 patients with IPF that had done oxygen therapy titration, defined as the minimal flow of oxygen needed to maintain minimum SpO_2_; (Peripheral Capillary Oxygen Saturation) of 96% standing. A directly proportional relation between the oxygen flow and the mortality was shown, independent of other factors as FVC and 6-MWT results. Other studies demonstrated that ambulatory oxygen improves exercise capacity in IPF patients [58]. In the patients already on oxygen therapy, titrating the flow rate of oxygen appropriately also significantly increased the walked distance [58].

Taking into account the previous mentioned, and considering the lack of studies supporting a survival benefit from ambulatory oxygen in IPF patients, the indications and use of oxygen therapy in IPF is extrapolated from the experience in COPD. The use of long term oxygen in those patients is well established, based mostly on two principal clinical trials performed in that group: the nocturnal oxygen therapy trial (NOTT) [63] and the Medical Council Research study (MRC) [64]. In fact, the current guidelines of oxygen therapy [65] recommend long term oxygen in stable COPD patients with resting PaO_2_ (partial pressure of Oxygen) <55 mmHg or resting PaO_2_ <60 mmHg with evidence of polycythemia (hematocrit greater than 55%), pulmonary hypertension or cor pulmonale. This well-accepted relationship between the use of long term ambulatory oxygen and improved health outcomes in COPD raises the possibility that it could have similar effects in ILD. Therefore, although there is no strong demonstrated evidence of survival benefit, the use of oxygen in patients with rest or nocturnal hypoxemia (SpO_2_ < 88%) is recommended by International Guidelines [39]. Also, it is recommended to perform a 6-MWT and titrate the flow rate oxygen in patients with exertional hypoxemia.

### 4.3. Support Therapies in Acute Life-Threating Hospitalizations

An acute life-threatening hypoxemia during the course of IPF occurs in a 5–10% of patients. This event may lead to consideration of invasive or non-invasive ventilatory support (NIV) as other new options as extracorporeal membrane oxygenation (ECMO) or high-flow nasal cannula (HFNC) oxygen.

In IPF hospitalizations, especially in acute exacerbations, oxygen therapy even in high concentrations is the main support therapy, given the significant hypoxia produced by an important mismatch of ventilation/perfusion ratio (V/Q) with severe diffusion impairment [66]. Due to this critical situation, most patients are placed in an intensive care unit (ICU) environment for close surveillance and control. HFNC oxygen therapy is gaining strength and may provide high flows and high FiO_2_ (Fraction of inspired oxygen) delivery. The therapeutic benefit includes reducing the rate and work of breathing and improving oxygenation. Also, the humidification improves secretion clearance and maintenance of mucosal integrity in a more comfortable way for the patient. One of the advantages versus NIV is that it requires less training and can be applied outside ICUs. Although it is a promising treatment, the specific role in ILD is yet unknown given the lack of specific studies, except case series reported. More studies are needed to evaluate the real benefit to guide an adequate indication of this therapy. Nowadays in clinical practice, it has been used depending on the availability and center experience. Mostly to avoid more invasive procedures in exacerbations while adjusting specific etiology treatment or waiting for lung transplant.

Regarding mechanical ventilation, is known that patients admitted to the intensive care unit with acute respiratory failure needing invasive ventilation have a very poor prognosis [57] with a high mortality of up to 90%. In fact, invasive mechanical ventilation has proved no improvement in the survival or prognosis of these patients [67]. The general recommendation is therefore avoiding invasive ventilator treatment except in some circumstances: for example, while waiting for lung transplantation.

In this context, NIV may be an appropriate option with an important role in improving dyspnea. Vianello et al. [68] performed a retrospective study analyzing 18 patients with IPF and respiratory failure treated with NIV in the ICU. The results showed a greater survival than previously reported data from patients treated with invasive ventilation. Furthermore, Gungor et al. [69] evaluated the use of NIV in IPF and other ILDs with better overall prognosis, although a higher mortality rate was observed in patients who needed continuous NIV. Despite the studies mentioned, NIV responsiveness does not seem to have an impact in disease-related prognosis. Also, it is important to have in mind that compliance of NIV in this situation is difficult because of poor patient-ventilator adaptation due to the intense breathlessness and high rate of breathing.

Another possible strategy is the use of ECMO or veno-venous ECMO in patients awaiting a lung transplant that suffer an acute worsening [70]. The first reported experience with ILD patients treated with ECMO was presented by Trudzinski et al. [71]. In this study, ECMO was used in 21 ILD patients (33% IPF) with severe respiratory failure considered to be a lung transplant candidate. Six patients (29%) treated with ECMO underwent lung transplant and two died on the waiting list. In this field, ECMO seems to be an acceptable bridge to transplant in patients with ILD eligible for lung transplantation. Another strategy introduced by this group was awake-ECMO, conceived as ECMO without intubation. Following this strategy, Fuehner et al. [72] compared the outcomes of awake-ECMO with standard mechanical ventilation, with an 80% survival at six months after lung transplantation in the awake-ECMO group. Thus, the awake-ECMO strategy may reduce ventilator-associated pneumonia, allow early rehabilitation and could prevent lung injury produced by the ventilator.

### 4.4. Lung Transplantation

IPF is a progressive disease with poor prognosis. In the past years, new pharmacological treatment has been developed with good results in slowing down the progression of the disease. Despite the pharmacological advance, it remains an incurable disease where lung transplant plays an important role and should be considered in some patients as it enhances survival. Determining the time to referral and register on the transplant list could be difficult. In order to advice physicians, in 2015 the International Society for Heart and Lung transplantation released guidelines on the selection of lung transplant candidates [73]. The guidelines define two important moments: referral and listing.

The timing of referral is wider and includes:
Histopathologic or radiographic evidence of usual interstitial pneumonitis or fibrosing non-specific interstitial pneumonitis, regardless of lung function.Abnormal lung function with FVC <80% predicted or DLCO <40% predicted.Any dyspnea or functional limitation attributable to lung disease.Any oxygen requirement, even if only during exertion.For inflammatory interstitial lung disease (ILD); failure to improve dyspnea, oxygen requirement, and/or lung function after a clinically indicated trial of medical therapy.

Regarding when is recommended to list these patients for transplant; the following situations were proposed:
Decline in FVC >10% during six months of follow-up.Decline in DLCO >15% during six months of follow-up.Desaturation to <88% or distance <250 m on 6 MWT or 450 m decline in 6 MWT distance over a 6-month period.Presence of pulmonary hypertension on right heart catheterization or 2-dimensional echocardiography.Hospitalization because of respiratory decline, pneumothorax, or acute exacerbation.

In addition to the above, it is also important to ensure adequate nutritional status and social support.

Another initiative to improve the selection of candidates for lung transplant was published in May 2005 by the United Network for Organ Sharing: the Lung Allocation Score (LAS). The LAS is nowadays a widely used method in many countries to identify the best candidates for transplant. The score is calculated using different measures of patient’s health that estimate survival probability and projected duration of survival with or without lung transplant. Since the implementation of LAS, a greater number of IPF patients received a lung transplant and the percentage of patients on the waiting list with restrictive lung disease has increased [74]. In fact, nowadays the second most common indication for lung transplant is interstitial lung disease (30%), mainly IPF (24.8%) [75].

Considering the type of procedure, both single and bilateral lung transplantations are performed in IPF patients and there is still a current debate on the election method. The lack of randomized trials to compare the both procedures does not allow showing a significant difference between single or bilateral transplantation, although some retrospective studies point out that survival could improve in patients with a bilateral lung transplant [76,77].

In one study, patients with IPF who received a lung transplant showed a 75% reduction in risk of death compared with patients who remained on the waiting list. Risk of mortality while waiting for lung transplantation is significantly greater for patients with IPF compared with cystic fibrosis and emphysema [57]. The overall median survival for all adult lung recipients is 6 years [75], and 4.5 years in IPF transplanted patients. The 3-month post-transplant mortality is 15% in patients with IPF, compared to 9% in patients with COPD. Age at transplant and comorbidities in IPF and COPD patients may be attributable to the difference in survival compared to other diagnostic groups [78].

According to the above mentioned and international guidelines, patients with IPF should be considered for lung transplant at the time of diagnosis to achieve an early referral to evaluation.

### 4.5. Pulmonary Rehabilitation

Pulmonary rehabilitation (PR) is defined by the American Thoracic Society (ATS) and the European Respiratory Society (ERS) as a “comprehensive intervention based on a thorough patient assessment followed by patient-tailored therapies that include, but are not limited to, exercise training, education, and behavior change, designed to improve the physical and psychological condition of people with chronic respiratory disease and to promote the long-term adherence to health-enhancing behaviors” [79].

In ILD, it is well known that exercise intolerance is an important feature due to its association with exertional dyspnea. Low physical activity is related to a deterioration in quality of life and poor survival [79]. In the case of IPF, many factors contribute to exercise limitation, such as impaired gas exchange with hypoxia, altered respiratory mechanics, fatigue and loss of muscular mass. For all displayed, pulmonary rehabilitation may be beneficial in those patients. However, it is also important to mention that patients with advanced disease severe and physical disability might not be able to perform the program at full intensity. This means that they will not achieve the maximum benefit. Hence, early referral of IPF patients for pulmonary rehabilitation (PR) may be appropriate.

The duration of PR programs varies between 6–12 weeks but is not well established because of the yet few patients included in the studies performed with ILD and IPF patients. Likewise, rehabilitation programs are not standardized and vary based on local practice. However, those programs usually perform muscular training with strength and resistance exercises and also respiratory reeducation through control of respiration techniques and diaphragmatic effort to prevent tachypnea, anxiety and to improve gas exchange.

Until recently, most respiratory rehabilitation studies had been conducted in COPD. Nowadays, new evidence suggests a short-term benefit of those programs in ILD patients. For example, a Cochrane review in 2014 showed that pulmonary rehabilitation reduced dyspnea, increased walked distance (average difference of 44.34 m in ILD and 35.63 m in IPF) and improved quality of life [80]. Besides, some randomized controlled trials performed demonstrated short-term improvements in functional exercise tolerance, dyspnea and quality of life after pulmonary rehabilitation. Holland et al. [81] performed a trial in a cohort of 44 patients with ILD (including 25 patients with IPF), finding a greater benefit of PR in patients with less severe IPF, less exercise-induced oxygen desaturation and lower right ventricular systolic pressure. However, the magnitude of these benefits was smaller than the observed in COPD [79]. The Nishiyama et al. [82] study included 28 IPF patients. Of those, 13 participated in a ten weeks PR program showing an improvement in walking distance compared to control group (the median walked distance difference between groups was 46.3 m). Another study performed by Huppmann et al. [83] included 402 patients, 202 of those with IPF and a median FVC of 53%. The IPF patients presented a median 45 m improvement in 6-MWT. In this trail, the patients with greatest improvement were those with the worst baseline walk distances. More recently, Jackson et al. [84] performed a randomized trial in 21 IPF patients to determine the efficacy of pulmonary rehabilitation in improving 6-MWT, dyspnea and muscle strength. The patients were randomly assigned to a 3-month pulmonary rehabilitation program or to a control group. This small trial showed that patients who completed the program had an improvement of symptoms. In conclusion, PR in IPF has been associated with a statistically significant improvement in 6-min walk distance (the minimum clinical difference estimated is a 28 m improvement) and dyspnea [81]. The roles of maintenance exercise programs following the initial structured training remains unclear. 

Despite the aforementioned, pulmonary rehabilitation is weakly recommended in the international guidelines of the ATS/ERS/JRS/ALAT, as the long-term benefit of pulmonary rehabilitation remains unclear. Further large randomized control trials are needed to explore the best IPF-specific PR protocol, optimum duration and aftercare to assess long-term benefits.

### 4.6. Palliative Care and Physicological Support

The management of idiopathic pulmonary fibrosis (IPF) should be oriented to a multidisciplinary care approach, including control of symptoms and psychological care, as this disease affects the quality of life of the patient, as well as that of his family and caregivers.

The World Health Organization defines palliative care as the “approach that improves the quality of life of patients and families who face the problems associated with life threatening diseases, through the prevention and alleviation of suffering through the early identification and impeccable evaluation and treatment of pain and other physical, psychological and spiritual problems.” It is evident that this is the situation of many patients affected by IPF, given the evolution and progression of this disease.

Gómez-Batiste [85], identifies the steps to perform comprehensive and integrated care, starting with an assessment of needs from a multidimensional approach (symptoms, functional, nutritional, cognitive, emotional, social), thus reviewing both the functional status and prognosis of the disease and comorbidities, and adapting the treatment, and identifying the patient’s preferences and planning their decisions in advance. But, as mentioned in a recent and broad review by Kreuter et al. [86], the great challenge is to identify the patients with these needs and to define when is the moment to initiate a palliative intervention. In that review, it is also mentioned that some patients and caregivers prefer to delay discussions about end of life management, while many other patients value early discussions. There are different tools that help us identify them as the Prognostic Indicator Guidance of the Gold Standard Framework, the Scottish Palliative Care Indicator Tool, NECPAL CCOMS-ICO (A Spanish practical tool), or the National Hospice Organization Criteria. 

Depression and anxiety, related to the diagnosis and prognosis, is prevalent in patients with IPF. It has a significant impact on QOL. So, it is important to perform a psychological assessment regularly in these patients.

Few studies have evaluated the role of palliative care in IPF, and these show shortcomings in the control of symptoms, psychological care and the planning of last wills. It also reveals the need to develop a strategy for the evaluation, treatment and monitoring of the palliative needs of patients [86]. This will appear with the progression of the disease, with the aim of improving the quality of life and the control of symptoms. In this context, effective communication is crucial to assess and readjust patient’s therapies throughout the whole process [86].

There are studies suggesting that the implementation of a multidisciplinary care model, including palliative care, in IPF patients, is associated with a decrease in the number of admissions in the situation of last days, as well as in the consumption of health care resources in the last year of life [87]. Patients who wish to express a preference regarding use of life-supporting interventions (e.g., intubation or cardiopulmonary resuscitation), should be encouraged to engage in discussions with their surrogates and physicians so that there is a clear understanding of the patient’s goals [57]. For example, another aspect evaluated in IPF patients is the place to die. A study performed in USA showed that most of patients died in the hospital without a previous evaluation by the palliative care team. Also, there was no previous conversation about the end of life. In the cases with a previous palliative care contact, the evaluation was made on the last month of life [88,89]. A patient support group is helpful to encourage and support family involvement in planning and providing care to the extent desired by the patient.

In conclusion, the early inclusion of patients with IPF in an advanced phase in a multidisciplinary palliative care program helps to adequately control symptoms, psychological support and makes decisions regarding the end of life, with the aim of maintaining the quality of life during the whole process of the disease.

## 5. Conclusions

Management of IPF is a challenge for respiratory physicians due to the unpredictable course of the disease and the significant impact in QOL and survival. Beyond pharmacological treatment, a comprehensive approach includes non-pharmacological therapies. An example is the growing role and value of palliative care in patients with advance disease or poor symptoms control.

Regarding comorbidities, in this review we have provided evidence-based information to highlight the value of an early assessment and management.

Acute exacerbation remains the main complication of patients with IPF. In recent years, there has been a significant interest in this field but still there is no effective treatment and prevention remains the clue.

## Figures and Tables

**Figure 1 medsci-06-00059-f001:**
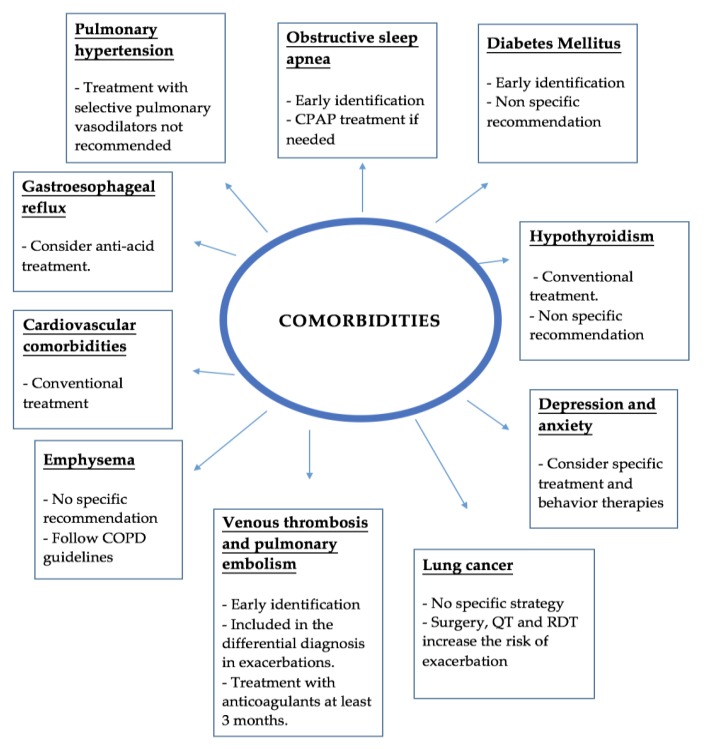
Summary of comorbidities and complications. CPAP—Continuous Positive Airway Pressure; COPD—Chronic Obstructive Pulmonary Disease; QT—Chemotherapy; RDT—Radiotherapy.

**Figure 2 medsci-06-00059-f002:**
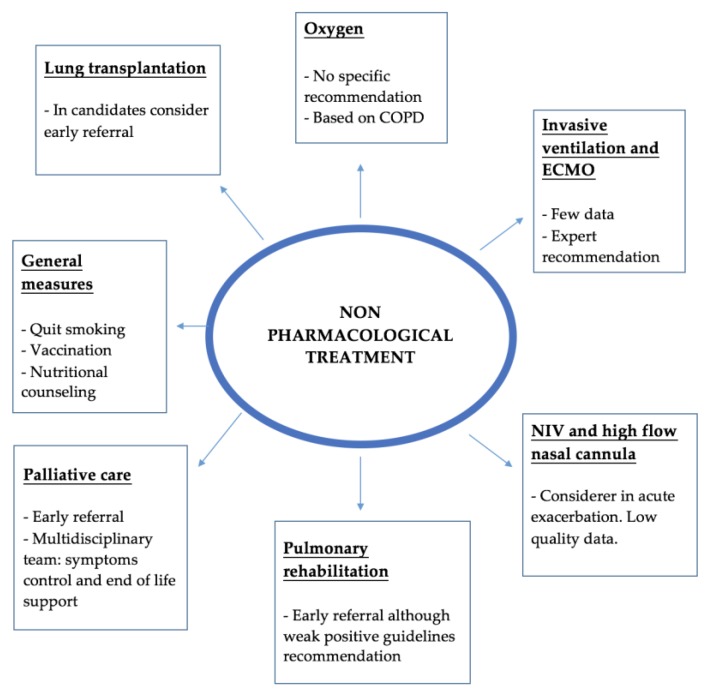
Summary of the main non-pharmacological treatment. ECMO—Extracorporeal membrane oxygenation; NIV—Non-invasive ventilation.

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
