# Peer review of "Comorbidities, Complications and Non-Pharmacologic Treatment in Idiopathic Pulmonary Fibrosis"

_medsci, 2018, doi:10.3390/medsci6030059_

Round 1

Reviewer 1 Report

The authors have undertaken an extensive review of the co-morbidities, complications and non-pharmacologic treatment in IPF.  The non-pharmacological treatments discussed are for the treatment of IPF rather than the co-morbidities and complications.

Major Comments

1. Grouping together complications and co-morbidities with the non-pharmacological treatments for IPF is confusing in places.  I suggest separating them so that the manuscript focuses mainly on the complications and co-morbidities with review of the literature to provide evidence or recommendations based upon best practice for treating these. This would improve the structure and flow of the manuscript.

2. The authors should consider other important co-morbidities of IPF such as increase risk of PE (haematological disorders), metabolic co-morbidities including diabetes mellitus and hypothyroidism, depression and anxiety.

3. Comments with regards to the specific complications and co-morbidities:

Pulmonary hypertension: although right heart catheterisation is the “gold standard” diagnostic test for pulmonary hypertension, it is not regularly performed in patients with IPF. The authors refer to 2011 ATS/ERS/JRS/ALAT guidelines for treatment recommendations, which were superceded by 2015 guidelines, which have a conditional recommendation against use of dual endothelin receptor antagonists and phosphodiesterase-5 inhibitor (sildenafil). There are no references provided to support the comments about the clinical trials with ambrisentan and riociguat.

Emphysema: line 83 – phenotype of FPI – should this be IPF?  The authors need to clarify the comment about lung transplantation as a treatment recommendation in CPFE section. There are many factors to consider for lung transplantation and it may not be a suitable treatment for patients with IPF including CPFE phenotype. Furthermore, although ASCEND and INPULSIS studies included patients with emphysema, the studies excluded those with significant obstructive airways disease based upon FEV1/FVC ratio.

OSA section: line 104 – OSA and IFP – should this be IPF?

Gastroesophageal reflux section: need to reference the clinical trials discussed (PANTHER/ACE/STEP/ASCEND/CAPACITY). None of these studies were designed to evaluate effect of anti-acid therapy on FVC decline.  This was retrospective analysis and is a mixed population due to the variety of treatments being used in the IPF clinical trials. Please can the authors amend this section.

AE-IPF section: this section needs clarification regarding the comments about lung transplantation and mechanical ventilation and ECMO. The comments are not in keeping with the updated 2015 guidelines. There are differences between countries with regards to criteria for lung transplant referral. Early lung transplantation referral should be considered for those patients who survive AE-IPF but may not be appropriate during AE-IPF if the patient is not already on a lung transplant waiting list. Similarly the guidelines do not recommend mechanical ventilation or ECMO for patients with IPF or AE-IPF as there is growing evidence that mechanical ventilation may precipitate an acute exacerbation. Limited evidence (based mainly on small retrospective studies) for ECMO as a bridge to lung transplantation in AE-IPF. The is a good discussion about mechanical ventilation and ECMO in the section on support therapies in acute life-threatening hospitalizations.

Furthermore, the risk of AE-IPF is not significantly reduced with nintedanib, the data are based upon a small number of patients who had AE-IPF in the clinical studies and are not statistically significant.

Oxygen section: need to reference Hook et al paper

Lung transplantation: the authors have provided accurate information about LAS. However, this has not been adopted by all countries worldwide. Line 369 “sing” should be single.

The conclusion section only discusses palliative care, please consider modifying this section as an overview with key messages from the manuscript.

Other comments

Throughout the manuscript there are many grammatical errors and inappropriate use of certain words, for instance page 1 line 24 “pretends” should be aims or intends and line 245 “pretends to reduce breathlessness” – may reduce…..

Multiple abbreviations are used in the manuscript, which need to be defined.

Ordering of the references in the text and reference list are not aligned in sections of the manuscript. For instance line 99, Lancaster et al is not reference 1 – this is a general review. The authors need to add the Lancaster reference as well as reviewing the text reference with the reference list.

Author Response

We are grateful to the reviewers because their helpful comments help to improve the manuscript. Following, we provide a detail description of the modifications on the text based on their suggestions: 

1.Grouping together complications and co-morbidities with the non-pharmacological treatments for IPF is confusing in places.  I suggest separating them so that the manuscript focuses mainly on the complications and co-morbidities with review of the literature to provide evidence or recommendations based upon best practice for treating these. This would improve the structure and flow of the manuscript.

Answer (A):Thank you for your comment and recommendation. Although we agree that the structure of the manuscript could be improved, the text is distributed following the journal's instructions to produce a chapter that would include comorbidities, complications and non-pharmacological treatment. 

2.The authors should consider other important co-morbidities of IPF such as increase risk of PE (haematological disorders), metabolic co-morbidities including diabetes mellitus and hypothyroidism, depression and anxiety. 

A:As recommended, we have added these comorbidities to the manuscript (see text).  

3Pulmonary hypertension: although right heart catheterisation is the “gold standard” diagnostic test for pulmonary hypertension, it is not regularly performed in patients with IPF. The authors refer to 2011 ATS/ERS/JRS/ALAT guidelines for treatment recommendations, which were superceded by 2015 guidelines, which have a conditional recommendation against use of dual endothelin receptor antagonists and phosphodiesterase-5 inhibitor (sildenafil). There are no references provided to support the comments about the clinical trials with ambrisentan and riociguat. 

A:. We thank the reviewer for this comment and apologize for the missing references. We have added the most recent recommendations made by the 2015 guidelines and specified references for the ambrisentan and riociguat studies. This is stated in the text as follow:

“...... evidenced on the right heart catheterization, the gold standard procedure for the diagnosis1, although it is not regularly performed in patients with idiopathic pulmonary fibrosis (IPF).”

“....In ATS/ERS/JRS/ALAT 2015 guidelines4authors made a conditional recommendation against the use Sildenafilo, a phosphodiesterase-5 inhibitor, based on a phase III randomized study (STEP-IPF)5  and a smaller study that showed no significant benefit on mortality or acute exacerbation in pooled analysis. In the same way, a conditional recommendation against the use of Bosentan or Macitentan, both dual endothelin receptor antagonists that showed no significant benefit in mortality although the data suggested an improvement in the composite outcome of mortality and disease progression. “

4.Emphysema: line 83 – phenotype of FPI – should this be IPF?  The authors need to clarify the comment about lung transplantation as a treatment recommendation in CPFE section. There are many factors to consider for lung transplantation and it may not be a suitable treatment for patients with IPF including CPFE phenotype. Furthermore, although ASCEND and INPULSIS studies included patients with emphysema, the studies excluded those with significant obstructive airways disease based upon FEV1/FVC ratio.

A:We thank the reviewer for this thoughtful comment. We agree that it is necessary to clarified that some patients were excluded from the analysis due to obstructive airway disease so we have modified the text as follow:

“....Both Pirfenidone (ASCEND) and Nintedanib (INPULSIS 1-2) trials included patients with emphysema. Despite the inclusion of emphysema patients, all the variability of patients with CPFE was not represented in those trials because patients with airway obstruction and lower DLCO were excluded. With the aforementioned, the efficiency of these antifibrotic drugs was maintained in the subgroup of patients with emphysema included in the trial”. 

Regarding lung transplantation, we included its evaluation as a general recommendation made in advanced respiratory patients, including the CPFE phenotype. If the reviewer feels it is essential to modify this section we will remove it. 

5.OSA section: line 104 – OSA and IFP – should this be IPF?

A: Changed in the text, thanks.

6.Gastroesophageal reflux section: need to reference the clinical trials discussed (PANTHER/ACE/STEP/ASCEND/CAPACITY). None of these studies were designed to evaluate effect of anti-acid therapy on FVC decline.  This was retrospective analysis and is a mixed population due to the variety of treatments being used in the IPF clinical trials. Please can the authors amend this section.

A:We thank the reviewer for this remark. As suggested we included an explanation about the retrospective nature of the mentioned analysis as well as the references to the clinical trials discussed.  

“…..In a retrospective analysis14of three randomized controlled trials (PANTHER15, ACE16, STEP5) anti-acid therapy was associated with lower decline in FVC and fewer acute exacerbations of IPF (AE-IPF) while in others, also retrospective,  analysis(ASCEND17, CAPACITY 004 and 00618) it was associated with an increased risk of infection19.Current IPF guidelines have proposed a conditional recommendation for anti-acid treatment based in previous findings, although scientific evidence is scarce in this topic20. For that reason, randomized prospective clinical trialsare now warranted to further study this therapy in this population.”

7.AE-IPF section: this section needs clarification regarding the comments about lung transplantation and mechanical ventilation and ECMO. The comments are not in keeping with the updated 2015 guidelines. There are differences between countries with regards to criteria for lung transplant referral. Early lung transplantation referral should be considered for those patients who survive AE-IPF but may not be appropriate during AE-IPF if the patient is not already on a lung transplant waiting list. Similarly the guidelines do not recommend mechanical ventilation or ECMO for patients with IPF or AE-IPF as there is growing evidence that mechanical ventilation may precipitate an acute exacerbation. Limited evidence (based mainly on small retrospective studies) for ECMO as a bridge to lung transplantation in AE-IPF. The is a good discussion about mechanical ventilation and ECMO in the section on support therapies in acute life-threatening hospitalizations.

A:  We agree with the reviewer about this issue. Accordingly, we have clarified the timing to lung transplantion evaluation and referral as well as the indication for mechanical ventilation and ECMO as bridge to the lung transplantation as follow: 

“-....Lung transplantation is proposed for appropriate patients with IPF, according to IPF guideline. Transplant eligible patients should be referred to a transplantation centre for early evaluation in the course of their disease, preferably before an episode of AE-IPF. Mechanical ventilation and extracorporeal membrane oxygenation are proposed as bridge for transplantation candidates.”

8.Furthermore, the risk of AE-IPF is not significantly reduced with nintedanib, the data are based upon a small number of patients who had AE-IPF in the clinical studies and are not statistically significant  

A:Because of the exposed by the reviewer, we have specified that no statistically significant differences were seen in the aforementioned study.

“...Moreover, treatment with nintedanib is proposed to reduce the risk of developing an acute exacerbation although no statistical significant difference in the incidence rates of investigator reported acute exacerbations in the nintedanib and placebo groups were seen”. 

9.Oxygen section: need to reference Hook et al paper.

A:We apologize for the missing reference. As suggested we added the reference.

10.Lung transplantation: the authors have provided accurate information about LAS. However, this has not been adopted by all countries worldwideLine 369 “sing” should be single.

A:We thank the reviewer for this remark, to clarified this issue we added a brief comment to clarified that LAS is used in “many countries”. Also we corrected the word single.

“The LAS is nowadays a widely used method in many countries to identify the best candidates for transplant.”

11. The conclusion section only discusses palliative care, please consider modifying this section as an overview with key messages from the manuscript.

A:We thank the reviewer for this good observation. As requested, we added a final conclusion including all the topics of the manuscript. 

12.Throughout the manuscript there are many grammatical errors and inappropriate use of certain words, for instance page 1 line 24 “pretends” should be aims or intends and line 245 “pretends to reduce breathlessness” – may reduce….. . Multiple abbreviations are used in the manuscript, which need to be defined.  

A: Following this comment we have reviewed the editing.

13.Ordering of the references in the text and reference list are not aligned in sections of the manuscript. For instance line 99, Lancaster et al is not reference 1 – this is a general review. The authors need to add the Lancaster reference as well as reviewing the text reference with the reference list.

A:References are updated and modified. Thanks.

Reviewer 2 Report

The authors have performed a review that is well-organized, clear, relevant and well-written. I only have some comments to suggest to the authors and editor:

1. One of my main concerns is the diferentiation between comorbidities and complications. Comorbidity is the presence of a disorder that coexists with the main disease (in our case IPF). Complication is a consequence of the main disorder. In the review, the authors have preferred to include both in the same section because some disorders could be considered a comorbidity and a complication (lung cancer). However, acute exacerbation could not be considered as a comorbidity. I suggest the authors clearly differentiate  comorbidities and complications in different sections.

2. Pulmonary thromboemobolism is not included as a complication of IPF....

3. The conclusions section is included after the acknoledgments  and COI section. In addition, the conclusions included were only from the palliative section, but no conclusions of the whole manuscript were included. 

Minor:

Revise if the reference 6 is correctly cited in page 3 , line 96. 

Page 4. In the GERsection, refer the correspondent articles instead of reference 1 (a review)

PAge 5. Paragraph 6. Must be "it is" instead of "it's"

Page 10. Line 394. Must be "they will not" instead of "won't".

Author Response

We are grateful to the reviewers because their helpful comments help to improve the manuscript. Following, we provide a detail description of the modifications on the text based on their suggestions: 

1. One of my main concerns is the differentiation between comorbidities and complications. Comorbidity is the presence of a disorder that coexists with the main disease (in our case IPF). Complication is a consequence of the main disorder. In the review, the authors have preferred to include both in the same section because some disorders could be considered a comorbidity and a complication (lung cancer). However, acute exacerbation could not be considered as comorbidity. I suggest the authors clearly differentiate comorbidities and complications in different sections.

A:  Thank you for your recommendation. We agree with the view of the reviewer. Therefore we have differentiated comorbidities, complications and non pharmacological treatment in 3 different sections. We decided to include lung cancer and embolic events as comorbidities following the other publications and reviewer 1 comments. But we will be happy to include it in the “complication section” if is need it. 

2.Pulmonary thromboembolism is not included as a complication of IPF....

A:We thank the author for this comment. As suggested, we added pulmonary thromboembolism in the manuscript. But the “reviewer 1” suggested including it as a complication. Based on the literature, we decided to follow this suggestion but we will happy to place it in the “complication section” if need it.  

3.The conclusions section is included after the acknoledgments  and COI section. In addition, the conclusions included were only from the palliative section, but no conclusions of the whole manuscript were included.

A: We thank the reviewer for this good observation. As requested, we added a final conclusion including all the topics of the manuscript. We also apologize for the editing error in the acknowledgments and COI section.

4. Revise if the reference 6 is correctly cited in page 3 , line 96. 

A:Thank you for this comment, we have reviewed this reference and did not correspond to the one mentioned above and has therefore been amended.

5.Page 4. In the GER section, refer the correspondent articles instead of reference 1 (a review)A:As suggested, we have modified the references.  

6.Page 5. Paragraph 6. Must be "it is" instead of "it's" .Page 10. Line 394. Must be "they will not" instead of "won't".

A:Modified. Many thanks.

Round 2

Reviewer 2 Report

The authors have addressed my comments adequately. In my opinion, the manuscript is suitable of publication.